# A Latent Morphology Model for Open-Vocabulary Neural Machine Translation

**Duygu Ataman**[*]
University of Zürich
ataman@cl.uzh.ch

**Wilker Aziz**
University of Amsterdam
w.aziz@uva.nl

**Alexandra Birch**
University of Edinburgh
a.birch@ed.ac.uk

## Abstract

Translation into morphologically-rich languages challenges neural machine translation (NMT) models with extremely sparse vocabularies where atomic treatment of surface forms is unrealistic. This problem is typically addressed by either pre-processing words into subword units or performing translation directly at the level of characters. The former is based on word segmentation algorithms optimized using corpus-level statistics with no regard to the translation task. The latter learns directly from translation data but requires rather deep architectures. In this paper, we propose to translate words by modeling word formation through a hierarchical latent variable model which mimics the process of morphological inflection. Our model generates words one character at a time by composing two latent representations: a continuous one, aimed at capturing the lexical semantics, and a set of (approximately) discrete features, aimed at capturing the morphosyntactic function, which are shared among different surface forms. Our model achieves better accuracy in translation into three morphologically-rich languages than conventional open-vocabulary NMT methods, while also demonstrating a better generalization capacity under low to mid-resource settings.

## 1 Introduction

Neural machine translation (NMT) models are conventionally trained by maximizing the likelihood of generating the target side of a bilingual parallel corpus of observations one word at a time conditioned of their full observed context. NMT models must therefore learn distributed representations that accurately predict word forms in very diverse contexts, a process that is highly demanding in terms of training data as well as the network capacity. Under conditions of lexical sparsity, which includes both the case of unknown words and the case of known words occurring in surprising contexts, the model is likely to struggle. Such adverse conditions are typical of translation involving morphologically-rich languages, where any single root may lead to exponentially many different surface realizations depending on its syntactic context. Such highly productive processes of word formation lead to many word forms being rarely or ever observed with a particular set of morphosyntactic attributes. The standard approach to overcome this limitation is to pre-process words into subword units that are shared among words, which are, in principle, more reliable as they are observed more frequently in varying context (Sennrich et al., 2016; Wu et al., 2016). One drawback related to this approach, however, is that the estimation of the subword vocabulary relies on word segmentation methods optimized using corpus-dependent statistics, disregarding any linguistic notion of morphology and the translation objective. This often produces subword units that are semantically ambiguous as they might be used in far too many lexical and syntactic contexts (Ataman et al., 2017). Moreover, in this approach, a word form is then generated by prediction of multiple subword units, which makes generalizing to unseen word forms more difficult due to the possibility that a subword unit necessary to reconstruct a given word form may be unlikely in a given context. To alleviate the sub-optimal effects of using explicit segmentation and generalize better to new morphological forms, recent studies explored the idea of extending NMT to model translation directly at

---

[*]Work done while the first author was a doctoral student at the University of Trento and a visiting postgraduate student at the University of Edinburgh.

the level of characters (Kreutzer & Sokolov, 2018; Cherry et al., 2018), which, in turn, have demonstrated the requirement of using comparably deeper networks, as the network would then need to learn longer distance grammatical dependencies (Sennrich, 2017).

In this paper, we explore the benefits of explicitly modeling variation in surface forms of words using techniques from deep latent variable modeling in order to improve translation accuracy for low-resource and morphologically-rich languages. Latent variable models allow us to inject inductive biases relevant to the task, which, in our case, is word formation during translation. In order to formulate the process of morphological inflection, design a hierarchical latent variable model which translates words one character at a time based on word representations learned compositionally from sub-lexical components. In particular, for each word, our model generates two latent representations: i) a continuous-space dense vector aimed at capturing the lexical semantics of the word in a given context, and ii) a set of (approximately) discrete features aimed at capturing that word's morphosyntactic role in the sentence. We then see inflection as decoding a word form, one character at a time, from a learned composition of these two representations. By forcing the model to encode each word representation in terms of a more compact set of latent features, we encourage them to be shared across contexts and word forms, thus, facilitating generalization under sparse settings. We evaluate our method in translating English into three morphologically-rich languages each with a distinct morphological typology: Arabic, Czech and Turkish, and show that our model is able to obtain better translation accuracy and generalization capacity than conventional approaches to open-vocabulary NMT.

## 2  NEURAL MACHINE TRANSLATION

In this paper, we use recurrent NMT architectures based on the model developed by Bahdanau et al. (2014). The model essentially estimates the conditional probability of translating a source sequence $x = \langle x_1, x_2, \ldots x_m \rangle$ into a target sequence $y = \langle y_1, y_2, \ldots y_l \rangle$ via an exact factorization:

$$p(y|x, \theta) = \prod_{i=1}^{l} p(y_j | x, y_{<i}, \theta) \tag{1}$$

where $y_{<i}$ stands for the sequence preceding the $i$th target word. At each step of the sequence, a fixed neural network architecture maps its inputs, the source sentence and the target prefix, to the probability of the $i$th target word observation in context. In order to condition on the source sentence fully, this network employs an embedding layer and a bi-directional recurrent neural network (bi-RNN) based encoder. Conditioning on the target prefix $y_{<i}$ is implemented using a recurrent neural network (RNN) based decoder, and an attention mechanism which summarises the source sentence into a context vector $\mathbf{c}_i$ as a function of a given prefix (Luong et al., 2015). Given a parallel training set $\mathcal{D}$, the parameters $\theta$ of the network are estimated to attain a local minimum of the negative log-likelihood function $\mathcal{L}(\theta|\mathcal{D}) = -\sum_{x,y \sim \mathcal{D}} \log p(y|x, \theta)$ via stochastic gradient-based optimization (Bottou & Cun, 2004).

**Atomic parameterization**   estimates the probability of generating each target word $y_i$ in a single shot:

$$p(y_i | x, y_{<i}, \theta) = \frac{\exp(\mathbf{E}_{y_i}^{\top} \mathbf{h}_i)}{\sum_{e=1}^{v} \exp(\mathbf{E}_e^{\top} \mathbf{h}_i)} , \tag{2}$$

where $\mathbf{E} \in \mathbb{R}^{v \times d}$ is the target embedding matrix and the decoder output $\mathbf{h}_i \in \mathbb{R}^d$ represents $x$ and $y_{<i}$. Clearly, the size $v$ of the target vocabulary plays an important role in determining the complexity of the model, which creates an important bottleneck when translating into low-resource and morphologically-rich languages due to the sparsity in the lexical distribution.

Recent studies approached this problem by performing NMT with *subword* units, a popular one of which is based on the Byte-Pair Encoding algorithm (BPE; Sennrich et al., 2016), which finds the optimal description of a corpus vocabulary by iteratively merging the most frequent character sequences. Atomic parameterization could also be used to model translation at the level of characters, which is found to be advantageous in generalizing to morphological variations (Cherry et al., 2018).

**Hierarchical paramaterization** further factorizes the probability of a target word in context:

$$p(y_i|x, y_{<i}, \theta) = \prod_{j=1}^{l_i} p(y_{i,j}|x, y_{<i}, y_{i,<j}, \theta) \tag{3}$$

where the $i$th word $y_i = \langle y_{i,1}, \ldots, y_{i,l_i} \rangle$ is seen as a sequence of $l_i$ characters. Generation follows one character at a time, each with probability computed by a fixed neural network architecture with varying inputs, namely, the source sentence $x$, the target prefix $y_{<i}$, and the prefix $y_{i,<j}$ of characters already generated for that word. In this case there are two recurrent cells, one updated at the boundary of each token, much like in the standard case, and another updated at the character level. Luong & Manning (2016) propose hierarchical parameterization to compute the probability $p(y_i|x, y_{<i}, \theta)$ for unknown words, while for known words they use the atomic parameterization. In this paper, we use the hierarchical parameterization method for generating *all* target words, where we also augment the input embedding layer with a character-level bi-RNN, which computes each word representation $\mathbf{y}_i$ as a composition of the embeddings of their characters (Ling et al., 2015).

## 3 A LATENT MORPHOLOGY MODEL (LMM) FOR LEARNING WORD REPRESENTATIONS

The application of a hierarchical structure for learning word representations in language modeling (Vania & Lopez, 2017) or semantic role labeling (Sahin & Steedman, 2018) have shown that such representations encode many cues about the morphological features of words by establishing a mapping between phonetic units and lexical context. Although it can provide an alternative solution to open-vocabulary NMT by potentially alleviating the need for subword segmentation, the quality of word representations learned by an hierarchical model is still highly dependant on the amount of observations (Sahin & Steedman, 2018; Ataman et al., 2019), since the training data is essential in properly modeling the lexical distribution. On the other hand, the process of word formation, particularly morphological inflection, has many properties that remain universal across languages, where a word is typically composed of a *lemma*, representing its lexical semantics, and a distinct combination of categorical *inflectional features* expressing the word's syntactic role in the phrase or sentence. In this paper, we propose to manipulate this universal structure in order to enforce an inductive bias on the prior distribution of words and allow the hierarchical parameterization model in properly learning lexical representations under conditions of data sparsity.

### 3.1 GENERATIVE MODEL

Our generative LMM for NMT formulates word formation in terms of a stochastic process, where each word is generated one character at a time by composing two latent representations: a continuous vector aimed at capturing the lemma, and a set of sparse features aimed at capturing the inflectional features. The motivation for using a stochastic model is twofold. First, deterministic models are by definition *unimodal*: when presented with the same input (the same context) they always produce the same output. When we model the word formation process, it is reasonable to expect a larger degree of ambiguity, that is, for the same context (*e.g.* a noun prefix), we may continue by inflecting the word differently depending on the (latent) mode of operation we are at (*e.g.* generating nominative, accusative or dative noun). Second, in stochastic models, the choice of distribution gives us a mechanism to favour a particular type of representation. In our case, we use *sparse* distributions for inflectional features to accommodate the fact that morphosyntactic features are discrete in nature. Our latent variable model is an instance of a variational auto-encoder (VAE; Kingma & Welling, 2013) inspired by the model of Zhou & Neubig (2017) for morphological reinflection.

Generation of the $i$th word starts by sampling a Gaussian-distributed representation in context. This requires predicting the Gaussian location $\mathbf{u}_i$ and scale $\mathbf{s}_i$ vectors,[1]

$$Z_i|x, y_{<i} \sim \mathcal{N}(\mathbf{u}_i, \mathrm{diag}(\mathbf{s}_i \odot \mathbf{s}_i))$$
$$\mathbf{u}_i = \mathrm{dense}(\mathbf{h}_i; \theta_{\mathrm{u}}) \tag{4}$$
$$\mathbf{s}_i = \zeta(\mathrm{dense}(\mathbf{h}_i; \theta_{\mathrm{s}}))$$

---

[1]**Notation** We use capital Roman letters for random variables (and lowercase letters for assignments). Boldface Roman letters are reserved for neural network output vectors, and $\odot$ stands for elementwise multiplication. Finally, we denote typical neural network layers as $\mathrm{layer}(\mathrm{inputs}; \mathrm{parameters})$.

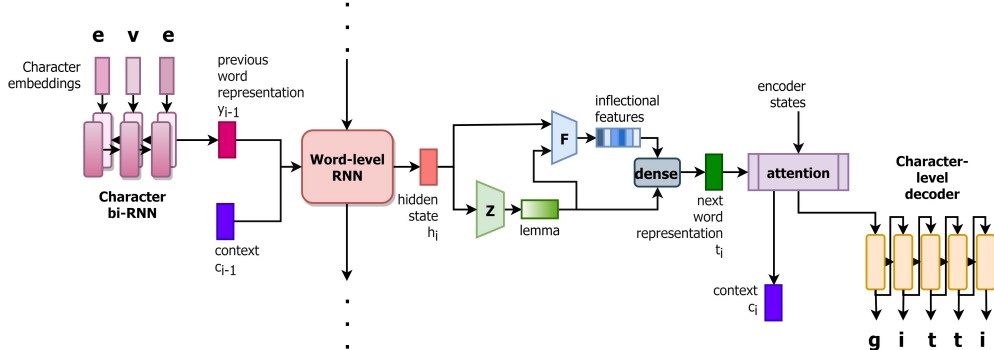

Figure 1: LMM for computing word representations while translating the sentence '... *went home*' into Turkish ('*eve*-(**to**)**home** *gitti*(**he/she/it**)**went**'). The character-level decoder is initialized with the attentional vector $\mathbf{h}_i$ computed by the attention mechanism using current context $\mathbf{c}_i$ and the word representation $\mathbf{t}_i$ as in Luong & Manning (2016).

where prediction of the location (in $\mathbb{R}^d$) and scale (in $\mathbb{R}^d_{>0}$) from the word-level decoder hidden state $\mathbf{h}_i$ (which represents $x$ and $y_{<i}$) is performed by two dense layers, and the scale values are ensured to be positive with the softplus ($\zeta$) activation.[2]

Generation proceeds by then sampling a $K$-dimensional vector $f_i$ of sparse scalar features (see §3.2) conditioned on the source $x$, the target prefix $y_{<i}$, and the sampled lemma $z_i$. We model sampling of $f_i$ conditioned on $z_i$ in order to capture the insight that inflectional transformations typically depend on the category of a lemma. Having sampled $f_i$ and $z_i$, the representation of the $i$th target word is computed by a transformation of $z_i$ and $f_i$, i.e. $\mathbf{t}_i = \mathrm{dense}([z_i, f_i]; \theta_{\mathrm{comp}})$.

As shown in Figure 1, our model generates each word character by character auto-regressively by conditioning on the word representation $\mathbf{t}_i$ predicted by the LMM, the current context $\mathbf{c}_i$, and the previously generated characters following the hierarchical parameterization.[3] See Algorithm 1 for details on generation.

**Input:** model parameters $\theta$, latent lemma $z_i$, latent morphological attributes $f_i$, observed
      character sequence $\langle y_{i,1}, \ldots, y_{i,l_i} \rangle$ if training or placeholders if test, decoder state $\mathbf{h}_i$,
      and context vector $\mathbf{c}_i$
**Result:** updated decoder hidden state, prediction (a word), probability of prediction (for loss)
initialization;
$\mathbf{t}_i = \mathrm{dense}([z_i, f_i]; \theta_{\mathrm{comp}})$ ;
initialize char-rnn with a projection of $[\mathbf{t}_i, \mathbf{c}_i]$;
**for** $j < l_i$ *and* $j < max$ **do**
    compute output layer from char-rnn state ;
    **if** *training* **then**
        set prediction to observation $y_{i,j}$ ;
    **else**
        set prediction to $\arg\max$ of output softmax layer ;
    **end**
    assess log-probability of prediction ;
    update word-level RNN decoder with prediction;
**end**
**Algorithm 1:** Word generation: in training the word is observed, thus we only update the decoder and assess the probability of the observation, in test, we use mean values of the distributions to represent most likely values for $z$ and $f$ and populate predictions with beam-search.

---

[2]In practice, we sample $z_i$ via a reparameterization in terms of a fixed Gaussian, namely, $z_i = \mathbf{u}_i + \epsilon_i \odot \mathbf{s}_i$ for $\epsilon_i \sim \mathcal{N}(0, I_d)$. This is known as the *reparameterization trick* (Kingma & Welling, 2013), which allows back-propagation through stochastic units (Rezende et al., 2014).

[3]Formally, because the decoder is an RNN, we are also conditioning on $z_{<i}$ and $f_{<i}$. We omit this dependence to avoid clutter.

## 3.2 SPARSE FEATURES

Since each target word $y_i$ may have multiple inflectional features, ideally, we would like $f_i$ to be $K$ feature indicators, which could be achieved by sampling from $K$ independent Bernoulli distributions parameterized in context. The problem with this approach is that sampling Bernoulli outcomes is non-differentiable, thus, their training requires gradient estimation via REINFORCE (Williams, 1992) and sophisticated variance reduction techniques. An alternative approach that has recently become popular is to use relaxations such as the Concrete distribution or Gumbel-Softmax (Maddison et al., 2017; Jang et al., 2017) in combination with the straight-through estimator (ST; Bengio et al., 2013). This is based on the idea of relaxing the discrete variable from taking on samples in the discrete set $\{0, 1\}$ to taking on samples in the continuous set $(0, 1)$ using a distribution for which a reparameterization exists (*e.g.* Gumbel). Then, a non-differentiable activation (*e.g.* a threshold function) maps continuous outcomes to discrete ones. ST simply ignores the discontinuous activation in the backward pass, *i.e.* it assumes the Jacobian is the identity matrix. This does lead to biased estimates of the gradient of the loss, which is in conflict with the requirements behind stochastic optimization (Robbins & Monro, 1951).

An alternative presented by Louizos et al. (2018) achieves a different compromise, it gets rid of bias at the cost of mixing both sparse and dense outcomes. The idea is to obtain a continuous sample $c \in (0, 1)$ from a distribution for which a reparameterization exists and stretch it to a continuous support $(l, r) \supset (0, 1)$ using a simple linear transformation $s = l + (r - l)c$. A rectifier is then employed to map the negative outcomes to 0 and the positive outcomes larger than one to 1, *i.e.* $f = \min(1, \max(0, s))$. The rectifier is only non-differentiable at $s = 0$ and at $s = 1$, however, because the stretched variable $s$ is sampled from a *continuous* distribution, the chance of sampling $s = 0$ and $s = 1$ is essentially 0. This stretched-and-rectified distribution allows: i) the sampling procedure to become differentiable with respect to the parameters of the distribution, ii) to sample sparse outcomes with an unbiased estimator, and iii) to calculate the probability of sampling $f = 0$ and $f = 1$ in closed form as a function of the parameters of the underlying distribution, which corresponds to the probability of sampling $s < 0$ and $s > 1$, respectively.

In their paper, Louizos et al. (2018) used the BinaryConcrete (or Gumbel-Sigmoid) as the underlying continuous distribution, the sparsity of which is controlled via a temperature parameter. However, in our study, we found this parameter difficult to predict, since it is very hard to allow a neural network to control its value without unstable gradient updates. Instead, we opt for a slight variant by Bastings et al. (2019) based on the Kumaraswamy distribution (Kumaraswamy, 1980), a two-parameters distribution that closely resembles a Beta distribution and is sparse whenever its (strictly positive) parameters are between 0 and 1. In the context of text classification, Bastings et al. (2019) shows this stretch-and-rectify technique to work better than methods based on REINFORCE.

For each token $y_i$, we sample $K$ independent Kumaraswamy variables in context,

$$
\begin{aligned}
C_{i,k}|x, y_{<i}, z_i &\sim \text{Kuma}(a_{i,k}, b_{i,k}) \quad k = 1, \ldots, K \\
[\mathbf{a}_i, \mathbf{b}_i] &= \zeta(\text{dense}([z_i, \mathbf{h}_i]; \theta_{\text{ab}}))
\end{aligned}
\tag{5}
$$

which makes a continuous random vector $c_i$ in the support $(0, 1)^K$.[4] We then stretch-and-rectify the samples via $f_{i,k} = \min(1, \max(0, l - (r - l)c_{i,k}))$ making $f_i$ a random vector in the support $[0, 1]^K$.[5] The probability that $f_{i,k}$ is exactly 0 is

$$
\pi_{i,k}^{\{0\}} = \int_0^{\frac{-l}{r-l}} \text{Kuma}(c|a_{i,k}, b_{i,k}) \mathrm{d}c
\tag{6a}
$$

and the probability that $f_{i,k}$ is exactly 1 is

$$
\pi_{i,k}^{\{1\}} = 1 - \int_0^{\frac{1-l}{r-l}} \text{Kuma}(c|a_{i,k}, b_{i,k}) \mathrm{d}c
\tag{6b}
$$

and therefore the complement

$$
\pi_{i,k}^{(0,1)} = 1 - \pi_{i,k}^{\{0\}} - \pi_{i,k}^{\{1\}}
\tag{6c}
$$

---

[4]In practice we sample $c_{i,k}$ via a reparameterization of a fixed uniform variable, namely, $c_{i,k} = (1 - (1 - \varepsilon_{i,k})^{1/b_{i,k}})^{1/a_{i,k}}$ where $\varepsilon_{i,k} \sim \mathcal{U}(0, 1)$, which much like the Gaussian reparameterization enables back-propagation through samples (Nalisnick & Smyth, 2016).

[5]We use $l = -0.1$ and $r = 1.1$. Figure 2 in the appendix illustrates different instances of this distribution.

is the probability that $f_{i,k}$ be any continuous value in the open set $(0, 1)$. In §3.4, we will derive regularizers based on $\pi_{i,k}^{(0,1)}$ to promote sparse outcomes to be sampled with large probability.

### 3.3 PARAMETER ESTIMATION

Parameter estimation of neural network models is typically done via maximum-likelihood estimation (MLE), where we approach a local minimum of the negative log-likelihood function via stochastic gradient descent with gradient computation automated by the back-propagation algorithm. Using the following shorthand notation:

$$\alpha(z_i) \triangleq p(z_i | x, y_{<i}, z_{<i}, f_{<i}, \theta) \tag{7a}$$

$$\beta(f_i) \triangleq \prod_{k=1}^{K} p(f_{i,k} | x, y_{<i}, z_{<i}, f_{<i}, z_i, \theta) \tag{7b}$$

$$\gamma(y_i) \triangleq \prod_{j=1}^{l_i} p(y_{i,j} | x, y_{<i}, z_{\leq i}, f_{\leq i}, y_{i,<j}, \theta) . \tag{7c}$$

The log-likelihood for a single data point can be formulated as:

$$\log p(y|x, \theta) = \log \int \prod_{i=1}^{l} \alpha(z_i)\beta(f_i)\gamma(y_i) \mathrm{d}z \mathrm{d}f \tag{8}$$

the computation of which is intractable. Instead, we resort to variational inference (VI; Jordan et al., 1999), where we optimize a lower-bound on the log-likelihood

$$\mathbb{E}_{q(z,f|x,y,\lambda)} \left[ \sum_{i=1}^{l} \log \frac{\alpha(z_i)\beta(f_i)\gamma(y_i)}{q(z, f|x, \lambda)} \right] \tag{9}$$

expressed with respect to an independently parameterized posterior approximation $q(z, f|x, y, \lambda)$. For as long as sampling from the posterior is tractable and can be performed via a reparameterization, we can rely on stochastic gradient-based optimization. In order to have a compact parameterization, we choose

$$q(z, f|x, y, \lambda) := \prod_{i=1}^{l} \alpha(z_i)\beta(f_i) . \tag{10}$$

This simplifies the lowerbound, which then takes the form of $l$ nested expectations, the $i$th of which is $\mathbb{E}_{\alpha(z_i)\beta(f_i)} [\log \gamma(y_i)]$. This is similar to the stochastic decoder of Schulz et al. (2018), though our approximate posterior is in fact, also our parameterized prior.[6] Although this objective does not particularly promote sparsity, we employ sparsity-inducing regularization techniques that will be discussed in the next section.

Concretely, for a given source sentence $x$, target prefix $y_{<i}$, and a latent sample $z_{\leq i}, f_{\leq i}$, we obtain a single-sample estimate of the loss by computing $\mathcal{L}_i(\theta) = -\log \gamma(y_i)$.

### 3.4 REGULARIZATION

In order to promote sparse distributions for the inflectional features, we apply a regularizer inspired by expected $L_0$ regularization (Louizos et al., 2018). Whereas $L_0$ is a penalty based on the number of non-zero outcomes, we design a penalty based on the expected number of *continuous outcomes*, which corresponds to $\pi_{i,k}^{(0,1)}$ as shown in Equation (6). For a given source sentence $x$, target prefix $y_{<i}$, and a latent sample $z_{<i}, f_{<i}$, we aggregate this penalty for each feature

$$\mathcal{R}_i(\theta) = \sum_{k=1}^{K} \pi_{i,k}^{(0,1)} \tag{11}$$

---

[6]This means we refrain from conditioning on the observation $y_i$ itself when sampling $z_i$ and $f_i$. Whereas this gives our posterior approximation access to less features than the true posterior would have, we do not employ a fixed uninformative prior, but rather an autoregressive network trained on the likelihood of generated word forms. This is a common approximation for latent variable models that employ autoregressive priors (Goyal et al., 2017).

and add it to the cost function with a positive weight $\rho$. The final loss of the NMT model is

$$\mathcal{L}(\theta|\mathcal{D}) = \sum_{x,y \sim \mathcal{D}} \sum_{i=1}^{|y|} \mathcal{L}_i(\theta) + \rho \mathcal{R}_i(\theta) . \tag{12}$$

## 3.5 PREDICTIONS

In our model, obtaining the conditional likelihood for predicting the most likely hypothesis requires marginalisation of the latent variables, which is intractable. An alternative approach is to heuristically search through the joint distribution,

$$\underset{y,z,f}{\arg\max} \ p(y,z,f|x) , \tag{13}$$

rather than the marginal, an approximation that has been referred to as *Viterbi decoding* (Smith, 2011). During beam search, we populate the beam with alternative target words, and for each prefix $y_{<i}$ in the beam, we resort to deterministically choosing the latent variables based on a single sample which we deem representative of their distributions, which is a common heuristic in VAEs for translation (Zhang et al., 2016; Schulz et al., 2018). For unimodal distributions, such as the Gaussian $p(z_i|x, y_{<i}, z_{<i}, f_{<i})$, we use the analytic mean, whereas for multimodal distributions, such as the Hard Kumaraswamy $p(f_i|x, y_{<i}, z_{\leq i}, f_{<i})$, we use the argmax.[7]

## 4 EVALUATION

### 4.1 MODELS

We evaluate our model by comparing it in machine translation against three baselines which constitute the conventional open-vocabulary NMT methods, including architectures using atomic parameterization either with subword units segmented with BPE (Sennrich et al., 2016) or characters, and the hierarchical parameterization method employed for generating all words in the output. We implement all architectures using Pytorch (Paszke et al., 2017) within the OpenNMT-py framework (Klein et al., 2017)[8].

### 4.2 DATA AND LANGUAGES

In order to evaluate our model we design two sets of experiments. The experiments in §4.4.1 aim to evaluate different methods under low-resource settings, for languages with different morphological typology. We model the machine translation task from English into three languages with distinct morphological characteristics: Arabic (*templatic*), Czech (*fusional*), and Turkish (*agglutinative*). We use the TED Talks corpora (Cettolo, 2012) for training the NMT models for these experiments. In §4.4.3, we conduct more experiments in Turkish to demonstrate the case of increased data sparsity using multi-domain training corpora, where we extend the training set using corpora from EU Bookshop (Skadiņš et al., 2014), Global Voices, Gnome, Tatoeba, Ubuntu (Tiedemann, 2012), KDE4 (Tiedemann, 2009), Open Subtitles (Lison & Tiedemann, 2016) and SETIMES (Tyers & Alperen, 2010)[9]. The statistical characteristics of the training sets are given in Tables 4 and 5. We use the official evaluation sets of the IWSLT[10] for validating and testing the accuracy of the models. In order to increase the number of unknown and rare words in the evaluation sets we measure accuracy on large test sets combining evaluation sets from many years (Table 6 presents the evaluation sets used for development and testing). The accuracy of each model output is measured using BLEU (Papineni et al., 2002) and chrF3 (Popović, 2015) metrics, whereas the significance of the improvements are computed using bootstrap hypothesis testing (Clark et al., 2011). In order to measure the accuracy in predicting the correct syntactic description of the references, we also compute BLEU

---

[7]We maximize across the three configurations of each feature, namely, $\max\{\pi_{i,k}^{\{0\}}, \pi_{i,k}^{\{1\}}, \pi_{i,k}^{(0,1)}\}$. If $\pi_{i,k}^{(0,1)}$ is highest, we return the mean of the underlying Kumaraswamy variable.

[8]Our software is available at: `https://github.com/d-ataman/lmm`

[9]The size of the resulting combined corpora is further reduced to filter out noise and reduce the computational cost of the experiments using data selection methods (Cuong & Simaan, 2014).

[10]The International Workshop on Spoken Language Translation

scores over the output sentences segmented using a morphological analyzer. We use the AlKhalil Morphosys (Boudchiche et al., 2017) for segmenting Arabic, Morphidata (Straková et al., 2014) for segmenting Czech and the morphological lexicon model of Oflazer (Oflazer & Kuruöz, 1994) and disambiguation tool of Sak (Sak et al., 2007) for segmenting Turkish sentences into sequences of lemmas and morphological features.

## 4.3 TRAINING SETTINGS

All models are implemented using gated recurrent units (GRU) (Cho et al., 2014), and have a single-layer bi-RNN encoder. The source sides of the data used for training all NMT models, and the target sides of the data used in training the subword-level NMT models are segmented using BPE with 16,000 merge rules. We implement all decoders using a comparable number of GRU parameters, including 3-layer stacked-GRU subword and character-level decoders, where the attention is computed after the $1^{st}$ layer (Barone et al., 2017) and a 3-layer hierarchical decoder which implements the attention mechanism after the $2^{nd}$ layer. All models use an embedding dimension and GRU size of 512. LMM uses the same hierarchical GRU architecture, where the middle layer is augmented using 4 multi-layer perceptrons with 256 hidden units. We use a lemma vector dimension of 150, 10 inflectional features (See §A.3 for experiments conducted to tune the feature dimensions) and set the regularization constant to $\rho = 0.4$. All models are trained using the Adam optimizer (Kinga & Ba, 2014) with a batch size of 100, dropout rate of 0.2, learning rate of 0.0004 and learning rate decay of 0.8, applied when the perplexity does not decrease at a given epoch.[11] Translations are generated with beam search with a beam size of 5, where the hierarchical models implement the hierarchical beam search algorithm (Ataman et al., 2019).

## 4.4 RESULTS

### 4.4.1 THE EFFECT OF MORPHOLOGICAL TYPOLOGY

The experiment results given in Table 1 show the performance of each model in translating English into Arabic, Czech and Turkish. In Turkish, the most sparse target language in our benchmark with rich agglutinative morphology, using character-based decoding shows to be more advantageous compared to the subword-level and hierarchical models, suggesting that increased granularity in the vocabulary units might aid in better learning accurate representations under conditions of high data sparsity. In Arabic, on the other hand, using a hierarchical decoding model shows to be advantageous compared to the subword and character-level models, as it might be useful in better learning syntactic dependencies. LMM obtains improvements of **0.51** and **0.30** BLEU points in Arabic and Turkish over the best performing baselines, respectively. The fact that our model can efficiently work in both Arabic and Turkish confirms that it can handle the generation of both concatenative and non-concatenative morphological transformations. The results in the English-to-Czech translation direction do not indicate a specific advantage of using either method for generating fusional morphology, where morphemes are already optimized at the surface level, although our model is still able to achieve translation accuracy comparable to the character and subword-level models.

### 4.4.2 PREDICTING UNSEEN WORDS

In addition to the general machine translation evaluation using automatic metrics, we perform a more focused statistical analysis to illustrate the performance of different methods in predicting unseen words by computing the average perplexity per character on the input sentences which contain out-of-vocabulary (OOV) words as suggested by Cotterell et al. (2018). We also analyze the outputs generated by each decoder in terms of the frequency of unknown words in each model output and the Kullback-Leibler (KL) divergence between the character trigram distributions of the references and outputs, which represents the coherence between the statistical distribution learned by each model and the reference translations.

Our analysis results generally confirm the advantage of increased granularity during the generation of unseen words, where the character-level decoder can generate a higher rate of unseen word forms and higher KL-divergence with the reference, suggesting superior ability in generalizing to new

---

[11]Perplexity is the exponentiated average negative log-likelihood per segment (BPE, or character) that a model assigns to a dataset. It corresponds to the model's average surprisal per time step.

| Model | AR | | | CS *(only in-domain)* | | | TR | | |
|---|---|---|---|---|---|---|---|---|---|
| | BLEU | t-BLEU | chrF3 | BLEU | t-BLEU | chrF3 | BLEU | t-BLEU | chrF3 |
| Subwords | 14.27 | 51.24 | 0.3927 | 16.60 | **54.22** | **0.4123** | 8.52 | 38.03 | 0.3763 |
| Char.s | 12.72 | 47.56 | 0.3804 | 16.94 | 52.80 | 0.4103 | 10.63 | 40.63 | 0.3810 |
| Hierarch. | 15.55 | 54.01 | 0.4154 | 16.79 | 48.27 | 0.4068 | 9.74 | 35.91 | 0.3771 |
| LMM | **16.06** | **55.97** | **0.4251** | **16.97** | 50.35 | 0.4095 | **10.93** | **45.47** | **0.3889** |

| Model | TR *(multi-domain)* | | |
|---|---|---|---|
| | BLEU | t-BLEU | chrF3 |
| Subwords | 10.42 | 42.65 | 0.3722 |
| Char.s | 8.94 | 37.12 | 0.3274 |
| Hierarch. | 10.35 | 40.54 | 0.3870 |
| LMM | **11.48** | **48.23** | **0.3939** |

Table 1: Above: Machine translation accuracy in Arabic (AR), Czech (CS) and Turkish (TR) in terms of BLEU and ChrF3 metrics as well as BLEU scores computed on the output sentences tagged with the morphological analyzer (t-BLEU) using in-domain training data. Below: The performance of models trained with multi-domain data. Best scores are in bold. All improvements over the baselines are statistically significant (p-value $< 0.05$).

output and not necessarily copying previous observations as the subword-level model, however, this advantage is more visible in Turkish and less in Czech or Arabic. The hierarchical decoder which performs the search at the level of words, on the other hand, behaves with less uncertainty in terms of the perplexity values although it cannot demonstrate the ability to generalize to new forms and neither can closely capture the actual distribution in the target language.

Due to its stochastic nature, our model yields higher perplexity values compared to the hierarchical model, whereas the values range between subword and character-based models, possibly finding an optimal level of granularity between the two solutions. The KL-divergence and OOV rates confirm that our model has the potential in better generalize to new word forms as well as different morphological typology.

| Model | AR | | | CS | | | TR | | |
|---|---|---|---|---|---|---|---|---|---|
| | OOV% | Ppl | KL-Div | OOV% | Ppl | KL-Div | OOV% | Ppl | KL-Div |
| Subwords | 1.75 | 2.84 | 12,871 | 2.39 | 2.62 | 8,954 | 3.54 | 2.78 | 17,342 |
| Char.s | 3.08 | 2.46 | 29,607 | 1.90 | 2.61 | 17,092 | 4.28 | 2.38 | 38,043 |
| Hierarch. | 1.96 | 2.59 | 15,064 | 0.87 | 2.65 | 29,022 | 1.53 | 2.46 | 68,743 |
| LMM | 3.78 | 2.68 | 9,892 | 2.4 | 2.71 | 14,296 | 4.89 | 2.59 | 38,930 |

Table 2: Percentage of out-of-vocabulary (OOV) words in the output, normalized perplexity measures (PPl) per characters and the KL divergence between the reference and outputs of systems trained with in-domain data on different language directions.

### 4.4.3 THE EFFECT OF DATA SIZE

Repeating the experiments in the English-to-Turkish translation direction by increasing the amount of training data with multi-domain corpora demonstrates a more challenging case, where there is a greater possibility of observing new words in varying context, either in the form of morphological inflections due to previously unobserved syntactic conditions, or a larger vocabulary extended with terminology from different domains. In this experiment, the character-level model experiences a drop in performance and its accuracy is much lower than the subword-level one, suggesting that its capacity cannot cope with the increased amount of training data. Empirical results suggest that with increased capacity, character-level models carry the potential to reach comparable performance to subword-level models (Cherry et al., 2018). On the other hand, our model reaches a much larger improvement of **0.82** BLEU points over the subword-level and **2.54** BLEU points over the character-level decoders, suggesting that it could make use of the increased amount of observations for improving the translation performance, which possibly aid the morphology model in becoming more accurate.

### 4.4.4 THE IMPACT OF INFLECTIONAL FEATURES

In order to understand whether the latent inflectional features in fact capture information about variations related to morphological transformations, we first try generating different surface forms of the same lemma by sampling a lemma vector with LMM for the input word *'go'* and generating outputs using the fixed lemma vector and assigning different values to the inflectional features. In the second experiment, we assess the impact of the inflectional features by setting all features $f$ to 0 and translating a set of English sentences with varying inflected forms in Turkish. Table 3 presents different sets of feature values and the corresponding outputs generated by the decoder and the outputs generated with or without the inflectional component.

| Features | Output | English Translation |
|---|---|---|
| [1,1,1,1,1,1,1,1,1,1] | git | go *(informal)* |
| [0,1,1,1,1,1,1,1,1,1] | *'a* git | *to* go |
| [0,1,0,1,1,1,1,1,1,1] | *'da* git | *at* go |
| [0,0,0,1,1,0,0,1,1,0] | gidin | go *(formal)* |
| [1,1,0,0,0,0,1,0,1,1] | gitmek | to go *(infinitive)* |
| [0,0,1,0,0,0,0,0,0,1] | gidiyor | (he/she/it is) going |
| [0,0,0,0,0,0,0,0,1,0] | gidip | *by* going *(gerund)* |
| [0,0,1,1,0,0,1,0,1,0] | gidiyoruz | (we are) going |

| Input | Output with $f$ | Output without $f$ |
|---|---|---|
| he went home. | ev**e** gitti. | ev**e** gitti. |
| he came from home. | ev**den** geldi. | ev**e** geldi. |
| it is good to be home. | ev**de** olmak iyi. | ev**de** olmak iyi. |
| his home has red walls. | ev**inde** kırmızı duvar**lar** var. | ev**de** kırmızı duvar var. |

Table 3: Above: Outputs of LMM based on the lemma *'git'* (*'go'*) and different sets of inflectional features. Below: Examples of predicting inflections in context with or without using features.

The model generates different surface forms for different sets of features, confirming that the latent variables represent morphological features related to the infinitive form of the verb, as well as its formality conditions, prepositions, person, number and tense. Decoding the set of sentences given in the second experiment LMM always generates the correct inflectional form, although when the feature values are set to 0 the model omits some inflectional features in the output, suggesting that despite partially relying on the source-side context, it still encodes important information for generating correct surface forms in the inflectional features.

## 5 CONCLUSION

In this paper we presented a novel decoding architecture for NMT employing a hierarchical latent variable model to promote sparsity in lexical representations, which demonstrated promising application for morphologically-rich and low-resource languages. Our model generates words one character at a time by composing two latent features representing their lemmas and inflectional features. We evaluate our model against conventional open-vocabulary NMT solutions such as subword and character-level decoding methods in translationg English into three morphologically-rich languages with different morphological typologies under low to mid-resource settings. Our results show that our model can significantly outperform subword-level NMT models, whereas demonstrates better capacity than character-level models in coping with increased amounts of data sparsity. We also conduct ablation studies on the impact of feature variations to the predictions, which prove that despite being completely unsupervised, our model can in fact manage to learn morphosyntactic information and make use of it to generalize to different surface forms of words.

## 6 ACKNOWLEDGMENTS

The authors would like to thank Marcello Federico, Orhan Fırat, Adam Lopez, Graham Neubig, Akash Srivastava and Clara Vania for their feedback and suggestions. This project received funding from the European Union's Horizon 2020 research and innovation programme under grant agreements 825299 (GoURMET) and 688139 (SUMMA).

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

# A APPENDIX

## A.1 THE STATISTICAL CHARACTERISTICS OF EXPERIMENTAL DATA

| Language Pair | # sentences | # tokens | | # types | |
|---|---|---|---|---|---|
| | | Source | Target | Source | Target |
| English-Arabic | 238K | 5M | 4M | 120K | 220K |
| English-Czech | 118K | 2M | 2M | 50K | 118K |
| English-Turkish | 136K | 2M | 3M | 53K | 171K |

Table 4: Training sets based on the TED Talks corpora (*M*: Million, *K*: Thousand).

| Language Pair | # sentences | # tokens | | # types | |
|---|---|---|---|---|---|
| | | Source | Target | Source | Target |
| English-Turkish | 434K | 8M | 6M | 135K | 373K |

Table 5: The multi-domain training set (*M*: Million, *K*: Thousand).

| Language | Data sets | | # sentences |
|---|---|---|---|
| English-Arabic | Development | dev2010, test2010 test2011, test2012 | 6K |
| | Testing | test2013, test2014 | 4K |
| English-Czech | Development | dev2010, test2010, test2011 | 3K |
| | Testing | test2012, test2013 | 3K |
| English-Turkish | Development | dev2010, test2010 | 3K |
| | Testing | test2011, test2012 | 3K |

Table 6: Development and testing sets (*K*: Thousand).

## A.2 THE KUMARASWAMY DISTRIBUTION

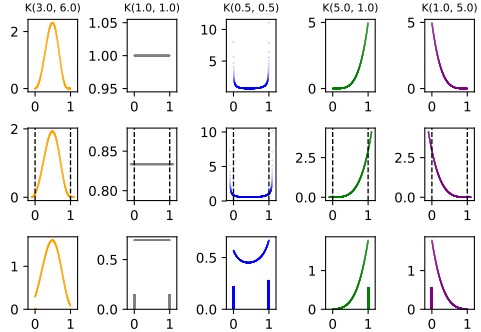

Figure 2: The top row shows the density function of the continuous base distribution over $(0, 1)$. The middle row shows the result of stretching it to include $0$ and $1$ in its support. The bottom row shows the result of rectification: probability mass under $(l, 0)$ collapses to $0$ and probability mass under $(1, r)$ collapses to $1$, which cause sparse outcomes to have non-zero mass. Varying the shape parameters $(a, b)$ of the underlying continuous distribution changes how much mass concentrates outside the support $(0, 1)$ in the stretched density, and hence the probability of sampling sparse outcomes.

## A.3 THE EFFECT OF FEATURE DIMENSIONS

We investigate the optimal lemma and inflectional feature sizes by measuring the accuracy in English-to-Turkish translation using different feature vector dimensions. The results given in Figure 3 show that gradually compressing the word representations computed by recurrent hidden states, with an original dimension of 512, from 500 to 100, leads to increased output accuracy, suggesting that encoding more compact representations might provide the model with a better generalization capability. Our results also show that using a feature dimension of 10 is sufficient in reaching the best accuracy.

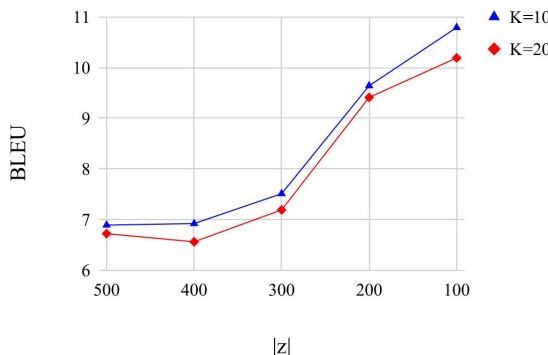

Figure 3: The effect of feature dimensions on translation accuracy in Turkish.

