# OpenReview forum: "A Latent Morphology Model for Open-Vocabulary Neural Machine Translation"
_ICLR.cc/2020/Conference — Accept (Spotlight)_

### Official Review · AnonReviewer2 · 2019-10-16
**Official Blind Review #2**

**Rating:** 6

**Review:**

This paper addresses the problem of translating into morphologically-rich languages, which suffers from the problem of sparse vocabularies and high numbers of rare and unseen words. In particular, current approaches such as subwords lack explicit notions of morphology and are obtained independently of the translation objective, while operating at the character-level renders the learning of long-distance dependencies more difficult.

More concretely, this paper models the generation of target words in a stochastic, hierarchical process, where the morphological features are modelled as latent variables. At each target word, the model samples a vector representing its lemma, followed by a k-dimensional latent inflectional features (e.g. nominative or accusative). To induce sparsity in the inflectional features, the paper uses a "stretch-and-rectify" distribution (Louizos et al., 2018) using the Kumaraswamy distribution. The paper further applies a sparsity-inducing regulariser to encourage the inflectional features to take discrete values of "0" or "1". Parameter estimation is done by optimising a lower-bound on the marginal log-likelihood. Experiments on translations into three morphologically-rich languages, English-{Arabic, Czech, Turkish}, indicate fairly small but consistent and statistically significant gains in BLEU. Further analysis indicates improved perplexity upper bound on rare words compared to subword-based baselines, along with somewhat interpretable latent inflectional features.

Overall, this paper proposes an elegant solution to an important problem, and yields statistically significant BLEU improvements over the baselines. I have listed some pros and cons below, with a recommendation of "Weak Accept". I would be willing to raise my scores assuming my concerns are sufficiently addressed.

Pros:
1. The proposed approach is easy-to-follow and explained clearly, and the paper is overall well-written.

2. Experiments are done on translation into three morphologically-rich languages with both concatenative and non-concatenative morphology, and results in small but mostly consistent BLEU improvements over the best subword, hierarchical, and character baselines for each language.

3. The proposed approach is general and can potentially be applied on top of other model architectures (e.g. Transformers) and other language generation problems, such as language modelling, summarisation, etc, although the experiments in the paper focus on NMT with a GRU-based architecture.

Cons:
1. It is still unclear how much computational overhead is introduced by the approach (in terms of both training and prediction times), and how scalable the approach is when applied to language pairs that have much bigger training sets. In Tables 4 and 5, it seems that the largest dataset is English-Turkish multi-domain (434K sentence pairs), which is fairly small compared to other datasets.

2. Regarding the feature variations result in section 4.4.4, it is hard to draw any conclusions about the latent inflectional features just based on varying the inflectional features at one position. I would suggest running the same experiment for multiple words at different positions, and see whether the same set of inflection features always results in similar inflections. This would better convince the reader that the latent inflectional features really are capturing useful morphological information, and that the target word generation process is appropriately controlled by the latent variable.

3. Since some of the experiments did not use the same standard splits (e.g. the multi-domain Turkish experiments), it would be nice to report how well external models (e.g. OpenNMT) would do in the authors' dataset split, to make sure that the reported numbers here are at least comparable to that. This would help ensure the credibility of the findings.

4. The effect of data size experiments (Section 4.4.2) for the character model seems somewhat counter-intuitive. Why would making the training set bigger (by incorporating data from a more diverse domain) make the character model worse? The explanation that "[the character model]'s capacity cannot cope with the increased amount of sparsity" does not seem satisfactory.  Why does adding more data result in increased sparsity? Furthermore, one can simply use more hidden units or deeper layers to mitigate this problem in the character-based model. Assuming the authors are controlling for model capacity, then the character model's hidden state can be made bigger (since the vocabulary size is lower for character models, thus the embedding layer and the softmax layer are by definition smaller than the subword-based model).

Minor suggestions:
1. Section 3.5 is not very clear; adding some figures or more explanation there would help understand how prediction is done.

2. It would be interesting to explore other potential metric for measuring morphological generalisation. For instance, at evaluation time, does the model predict words that it has never seen before (e.g. predicting "plays", even though the model has only seen "play" at training time) at a higher frequency than character/subword/hierarchical baselines? If yes, this would provide more evidence that the latent inflectional features are properly capturing morphological information.

**Experience Assessment:**

I have read many papers in this area.

**Review Assessment: Checking Correctness Of Derivations And Theory:**

I assessed the sensibility of the derivations and theory.

**Review Assessment: Checking Correctness Of Experiments:**

I carefully checked the experiments.

**Review Assessment: Thoroughness In Paper Reading:**

I read the paper at least twice and used my best judgement in assessing the paper.

---

### Official Review · AnonReviewer1 · 2019-10-21
**Official Blind Review #1**

**Rating:** 6

**Review:**


This paper proposes a method, Latent Morphology Model (LMM), for producing word representations for a (hierarchical) character-level decoder used in neural machine translation (NMT). The main motivation is to overcome vocabulary sparsity or highly inflectional languages such as Arabic, Czech, and Turkish (experimented in the paper). To model the morphological inflection, they decouple lemmas and inflection types into 2 latent variables (z and f) where f is enforced to be sparse (arguably mimic the process of the human). The literature review of NMT and the discussion on the potential advantage of morphology are concise. The proposed model is a variation of Luong & Manning (2016) and Schulz et al (2018) models, thus, their main contribution is an introduction of the latent morphological features to the decoder. The proposed LMM is trained by sampling z and f from prior directly, and the sparsity of the morphological features is encouraged by L0 of the feature vector (parameterized as independent Kumaraswamy variables). They perform the main empirical study using 3 languages by translating from English to justify the proposed LMM. Lastly, they provide a quantitative analysis of the perplexities of unseen words and a qualitative on words generated of a lemma with different feature vectors.

Overall, this paper could provide a novel insight into the role of modeling morphology as latent variables. However, the experiments and analyses do not sufficiently support the claim of mimicking the process of the inflection (besides the gain in performance). Some clarification would be appreciated.

In section 3, the model, to my understanding, is agnostic to the morphological inflection, except the sparsity regularization (i.e., it could model any transformation including changing the lemma itself). In addition, there is nothing in the training particularly specific about the morphology. For example, z and f are generated from only the prior whereas we could get more accurate posteriors from observing the word itself. Some background on the language might help motivate the choice of model. Are morphological inflections ambiguous? Are the morphology labels hard to obtain? I think more discussion on previous attempts to model morphology (e.g., Vylomova et al 2017, Passban et al 2018) will be very helpful to the readers.

For the experiments (section 4), the model with LMM is shown to consistently outperform character and hierarchical models. The multi-domain performance also shows the model's ability to tackle a larger vocabulary set. However, we cannot certainly conclude that the gain comes from LMM successfully modeling the inflections. A contrast of performance with less morphological languages might reveal some insight (unfortunately I do not have enough knowledge to recommend languages). Finally, the feature variation analysis is interesting, but we only see one lemma from one language. Further discussion such as the consistency of features and the types of morphology, or the similarity of the lemma vector across context, will be helpful.


Minor Questions:
1. Given a word, how ambiguous it is to determine the stem and the morphological type in the subject languages?
2. How do you compute char-level PPL of the subword model?
3. In 4.4.4, did you obtain z of `go` from just translating `go` without any context?



**Experience Assessment:**

I have read many papers in this area.

**Review Assessment: Checking Correctness Of Derivations And Theory:**

I assessed the sensibility of the derivations and theory.

**Review Assessment: Checking Correctness Of Experiments:**

I assessed the sensibility of the experiments.

**Review Assessment: Thoroughness In Paper Reading:**

I read the paper at least twice and used my best judgement in assessing the paper.

---

### Official Review · AnonReviewer3 · 2019-10-24
**Official Blind Review #3**

**Rating:** 8

**Review:**

This paper proposes to incorporate a latent model (in the form of a variational auto-encoder) in the decoding process of neural machine translation. The motivation is to capture morphology. Experiments on three language-pairs (English to Arabic, Czech, and Turkish) show promising improvements in translation accuracy (BLEU).

I am quite excited by this paper. So far there are not that many successful demonstrations of VAE in neural machine translation. The method is sound and interesting. The results show convincing improvements; some practioners may argue that reported BLEU gain (e.g. 0.8) is not impressive, but I think for a new model like this it is worthy.

Some suggestions or questions:

- One alternative evaluation metric that might be interesting is to lemmatize both translation outputs and reference, then compute BLEU. This will help distinguish whether the proposed method is improving by getting the morphological inflections correct, or whether it is improving across the board on various word types.

- Table 3 is interesting. If there is space, I would suggest more analysis along those directions, i.e. investigating what morphology is learned, what is in the latent spaces.

- Do you think the results will vary depending on decoder layer depth? I wonder if different kinds of latent spaces will be learned with different depth.

- Also a related question is how about varying the source input BPE merge operation? Again, it seems like these design choices might affect results, especially when dealing with morphology.


**Experience Assessment:**

I have published in this field for several years.

**Review Assessment: Checking Correctness Of Derivations And Theory:**

I assessed the sensibility of the derivations and theory.

**Review Assessment: Checking Correctness Of Experiments:**

I assessed the sensibility of the experiments.

**Review Assessment: Thoroughness In Paper Reading:**

I read the paper at least twice and used my best judgement in assessing the paper.

---

### Public Comment · ~Adithya_Renduchintala1 · 2019-10-16
**Choice of subword merge rules**

Thanks for posting on this interesting topic and the architecture proposed seems promising! Recently, some of our research has shown that for small datasets (such as TED) the BPE merge hyperparameter choice is very critical. We observe a 3-4 BLEU point difference simply by choosing a better BPE merge hyperparameter. Also, it seems like 32k for small data is generally a poor choice (see table 3 for details https://www.aclweb.org/anthology/W19-6620.pdf). Did you try other BPE choices?

---

### Decision · Program_Chairs · 2019-12-19

**Decision:**

Accept (Spotlight)

**Comment:**

This paper proposes a model for neural machine translation into morphologically rich languages by modeling word formation through a hierarchical latent variable model mimicking the process of morphological inflection. The model boils down to a VAE-like formulation with two latent representation: a continuous one (governed by a Gaussian) which captures lexical semantic aspects, and a discrete one (governed by the Kuma distribution) which captures the morphosyntactic function, shared among different surface forms. Even though the empirical improvements in terms of BLEU scores are fairly small, I find this a very elegant model which may foster interesting future research directions on latent models for NMT.

The reviewers had some concerns with some experimental details and model details that were properly addressed by the authors in their detailed response. In the discussion phase this alleviated the reviewers' concerns, which leads me to recommend acceptance. I urge the authors to follow the reviewer's recommendations to improve the final version of the paper.